# The Safe Use of ^125^I-Seeds as a Localization Technique in Breast Cancer during Pregnancy

**DOI:** 10.3390/cancers15123229

**Published:** 2023-06-17

**Authors:** Eva Heeling, Jeroen B. van de Kamer, Michelle Methorst, Annemarie Bruining, Mette van de Meent, Marie-Jeanne T. F. D. Vrancken Peeters, Christianne A. R. Lok, Iris M. C. van der Ploeg

**Affiliations:** 1Department of Surgical Oncology, Netherlands Cancer Institute Antoni van Leeuwenhoek, Plesmanlaan 121, 1066 CX Amsterdam, The Netherlands; e.heeling@nki.nl (E.H.); m.vrancken@nki.nl (M.-J.T.F.D.V.P.); 2Department of Radiation Oncology, Netherlands Cancer Institute Antoni van Leeuwenhoek, Plesmanlaan 121, 1066 CX Amsterdam, The Netherlands; j.vd.kamer@nki.nl; 3Department of Gynaecologic Oncology, Netherlands Cancer Institute Antoni van Leeuwenhoek, Plesmanlaan 121, 1066 CX Amsterdam, The Netherlands; m.methorst@nki.nl (M.M.); c.lok@nki.nl (C.A.R.L.); 4Department of Radiology, Netherlands Cancer Institute Antoni van Leeuwenhoek, Plesmanlaan 121, 1066 CX Amsterdam, The Netherlands; a.bruining@nki.nl; 5Department of Obstetrics, University Medical Center Utrecht, Heidelberglaan 100, 3584 CX Utrecht, The Netherlands; m.vandemeent@umcutrecht.nl

**Keywords:** breast cancer, pregnancy, breast-conserving surgery, iodine-125 seeds, radiation, fetus

## Abstract

**Simple Summary:**

The results of this multidisciplinary study can function as a clinician’s guide for determining the preferred surgical localization technique for breast-conserving surgery in pregnant women with breast cancer and its timing, without avoiding the use of a ^125^I-seed. The maximum exposure to the fetus remains well below 100 millisieverts (mSv), the threshold value for tissue damage caused by radiation. If the exposure should remain below 1 mSv, the implantation and surgical removal of the ^125^I-seed should occur within two weeks after a GA of 26 weeks, and within one week after a GA of 32 weeks. The contribution of an axillary seed is negligible (at maximum, 2.3% of the seed implanted in the breast).

**Abstract:**

Introduction: Some aspects of the treatment protocol for breast cancer during pregnancy (PrBC) have not been thoroughly studied. This study provides clarity regarding the safety of the use of ^125^I-seeds as a localization technique for breast-conserving surgery in patients with PrBC. Methods: To calculate the exposure to the fetus of one ^125^I-seed implanted in a breast tumor, we developed a model accounting for the decaying ^125^I-source, time to surgery, and the declining distance between the ^125^I-seed and the fetus. The primary outcome was the maximum cumulative fetal dose of radiation at consecutive gestational ages (GA). Results: The cumulative fetal dose remains below 1 mSv if a single ^125^I-seed is implanted at a GA of 26 weeks. After a GA of 26 weeks, the fetal dose can be at a maximum of 11.6 mSv. If surgery takes place within two weeks of implantation from a GA of 26 weeks, and one week above a GA of 32 weeks, the dose remains below 1 mSv. Conclusion: The use of ^125^I-seeds is safe in PrBC. The maximum fetal exposure remains well below the threshold of 100 mSv, and therefore, does not lead to an increased risk of fetal tissue damage. Still, we propose keeping the fetal dose as low as possible, preferably below 1 mSv.

## 1. Introduction

Breast cancer is the most common type of cancer that occurs during pregnancy (PrBC), with an incidence of approximately one in 3000 pregnancies [1,2,3,4,5,6]. The incidence of PrBC is likely to increase, presumably due to the trend of women becoming pregnant at an older age and breast cancer prevalence increasing with age [7,8]. Breast cancer during pregnancy is frequently diagnosed during the third trimester and often is a high-grade tumor with an aggressive clinical subtype [9,10,11,12,13,14,15]. The treatment of PrBC requires a multidisciplinary approach to determine the best diagnostic and therapeutic strategy for both maternal and fetal health [16]. In general, it is recommended to treat PrBC according to the same protocols as for non-pregnant patients, whenever feasible [17,18]. However, some aspects of the standard treatment protocol for breast cancer have not been thoroughly studied in pregnant patients.

Various localization methods can be used for primary breast-conserving surgery (BCS), or BCS after primary systemic treatment [19,20,21]. The use of iodine-125 (^125^I) seeds as a localization technique has grown in popularity in the past decade due to its high accuracy and low migration rate, and has rapidly replaced the former gold-standard wire-guided localization technique [19,22,23,24,25,26,27]. Moreover, re-excision rates are considerably lower when using ^125^I-seeds as a localization technique [28]. ^125^I-seeds are also used to mark tumor-positive axillary lymph nodes prior to neoadjuvant systemic treatment (NST), enabling their selective removal afterwards [29,30].

The ^125^I-seed, a double-layered titanium capsule containing the radionuclide ^125^I, emits a low dose of radiation that can be traced during surgery using a y-probe [23]. Breast cancer patients who are suitable for primary BCS often undergo ^125^I-seed placement during diagnostic work-up, around six weeks before surgery, usually after the pathology of the tumor is confirmed. In some hospitals, a hydromarker is placed during diagnostic work-up, and the ^125^I-seed marker is only inserted a few days before surgery. Breast cancer patients who are scheduled for NST, in whom the ^125^I-seed is placed shortly after diagnosis, maintain the seed during the entire treatment period, generally up to six to eight months, until it is removed during surgery [25]. The use of ^125^I-seeds in the treatment of PrBC remains a subject of controversy because of the fear that the level of radiation reaching the fetus in utero is harmful. Therefore, in PrBC patients, a non-radioactive marker, such as a hydromarker, is usually inserted during the diagnostic work-up, followed by an additional ^125^I-seed for localization before surgery.

The consequences of radiation for the fetus depend on the timing of the exposure in relation to the gestational age (GA) and the cumulative dose absorbed by the fetus [31]. According to the International Commission on Radiological Protection (ICRP), exposure to the fetus should not exceed the limit of 100 mSv in order to prevent the risk of fetal abnormalities [32,33,34,35,36]. Another possible negative consequence of radiation is the dose-dependent stochastic effect, namely the induction of cancer. To minimize this risk, the annual exposure of residents in the Netherlands is limited to 1 mSv in addition to normal background radiation [32,35]. For patients undergoing treatment in hospital, there is no exposure limitation, but the principle of “as low as reasonably achievable” (ALARA) is applicable. Although there is no legal exposure limit for the patient, including the fetus in the case of a pregnancy, extra efforts to keep the dose as low as reasonably achievable are indicated for pregnant women [37].

This study provides estimates of the exposure of a fetus at consecutive GAs to a ^125^I-seed in situ in a maternal breast tumor. The calculation accounts for (1) the decaying source, (2) the varying distance between the radioactive source in the breast and the growing fetus in the uterus, and (3) the duration between ^125^I-seed implantation and surgery. The aim of this study is to provide clarity regarding the safety of the use of ^125^I-seeds as a localization technique in BCS in cases of PrBC. This will equip healthcare providers and patients with indispensable information to make informed decisions regarding the use of ^125^I-seeds during pregnancy.

## 2. Materials and Methods

### 2.1. Study Design and Patient Population

We retrospectively included non-pregnant female patients with breast cancer, between the ages of 18 and 40 years, who had ^125^I-seeds implanted in breast tumors during their treatment at the Netherlands Cancer Institute—Antoni van Leeuwenhoek (NKI-AvL) in the past twenty years. Characteristics such as age, Body Mass Index (BMI), parity, and cup size were extracted from electronic patient files. We used archived 18F-fluorodeoxyglucose Positron Emission Tomography—Computed Tomography scans (PET-CT) performed during the course of treatment to measure anatomical distances from breast to uterus. The PET-CT scans were performed following local protocol, with women laying in prone position with breasts downwards, followed by a total body scan in supine position with feet first. We conducted measurements for our study on the total body scan in supine position. The distances between the ^125^I-seed in the breast tumor, the axillary lymph node and the uterus were measured to determine the range of distances. For the patients with the shortest distance between the ^125^I-seed and fetus, we also measured the distance from the axilla to the fundus, including the trajectory trough lung tissue. A dedicated breast cancer radiologist reviewed the measurements performed on PET-CT scans in two different directions.

To investigate whether the position of the breasts on the PET-CT scan in prone position was a good indicator of their position in a woman standing, we prospectively performed measurements on patients in the outpatient breast cancer clinic of the NKI-AvL to determine the distance between the nipple and os pubis symphysis. To estimate the decreasing distance between the breast (tumor) and the fetus for a woman sitting compared to standing, we determined the average decrease in distance between the breast and os pubis symphysis in the same patient group. The average decrease was used to determine the distance between standing and sitting in the dose calculations. These measurements were taken using a measuring tape with the bra on, and were approved by our internal review board.

### 2.2. Absorbed Radiation Dose Assessment

To calculate the maximum possible exposure of the fetus to the ^125^I-seed, we devised a model accounting for the decaying ^125^I-source (half-life: 59.4 days), the time between seed implantation and surgery, and the varying distance between seed and fetus during that period. The formulae and assumptions used to calculate the fetal dose are shown in Appendix A. McDonald’s rule (i.e., fundal height) (Section A.3, Table A1) provided the decrease in distance from the ^125^I-seed in the breast to the fetus in utero as a function of GA [36]. GA is measured in weeks, from the first day of a woman’s last menstrual period (LMP) to the current date. As the LMP is used to calculate the development of pregnancy, conception is at a GA of two weeks. Although McDonald’s rule states that in the first twelve weeks, the fetus resides below the level of the symphysis, we used the data obtained from our study population, yielding a shorter distance. This assumption results in a possible overestimation of fetal exposure, which is preferred from a safety point of view. The growth of the uterus was assumed to follow McDonald’s rule: 12 cm between a GA of, respectively, 12 and 16 weeks, with an increase of 4 cm per week until week 26. The National Institute of Standards and Technology of the United States of America provided data on the half-thickness values of fatty/muscle tissue (2.2 cm) [38]. For the calculations, we considered a minimum distance of 5 cm of muscle-like/fatty tissue to separate the fetus and the ^125^I-seed. This distance is occupied by, among others, the myometrium, amniotic membranes and fluid, muscle, intra-abdominal organs such as the liver, fatty tissue, and skin, all assigned a half-thickness value of 2.2 cm. From the study cohort, we noticed that the path of radiation from the ^125^I-seed to the fetus does not pass lung tissue, which has a lower attenuation coefficient compared to muscle and fatty tissue; the axillary ^125^I-seed does pass lung tissue. Using standard physics formulae considering the decaying source, distance, and attenuation by tissue, the estimated (maximum) exposure to the fetus (see Appendix A) was calculated. For the initial distance between the ^125^I-seed and the fetus, we used the smallest distance found in the cohort. The physical dose calculated (in milligrays (mGy)) was converted to the equivalent dose in mSv with a factor 0.35 mSv/mGy. In other words, one mGy equalizes 0.35 mSv. This is a cautious and very conservative estimate, based on the International Commission on Radiological Protection (ICRP)’s 116 values on isotropic exposure from an infinite plane [33].

To simulate all possible scenarios during the entire pregnancy, we performed calculations using one ^125^I-seed in situ, starting with seed insertion in week zero (conception) and surgery in week 1 and week 2, etc., until week 42. This was repeated for seed implantation in the week following conception (a GA of one), followed by surgery in the second week and third week, etc., until seed implantation at a GA of 41 and surgery at a GA of 42. Assuming that the ratio of sitting versus standing and/or lying per day is, respectively one third and two thirds, calculations were performed in this ratio using the shortest distances found [39,40]. For each calculation, the source activity used at implantation for one ^125^I-seed was 10 megabecquerels (0.358 μGy m^2^/h), measured at a one-meter distance. A dedicated medical physicist performed the calculations of the fictive levels of radiation reaching the fetus in utero.

### 2.3. Outcomes

The primary outcome was the cumulative dose of radiation absorbed by a fetus in utero at consecutive GAs with one ^125^I-seed in situ in the maternal breast. The secondary outcome was the maximum duration the seed can remain in situ until reaching a threshold of 1 mSv.

### 2.4. Sample Size and Statistical Analysis

We used a data saturation approach to determine the sample size. To present descriptive summary statistics, the mean and standard deviation (SD) and the 95% confidence interval (CI), or the median with the interquartile range (IQR), was calculated depending on whether there was a normal or non-normal distribution of the specific variable. Analyses were performed in IBM SPSS Statistics, version 27.0. Calculations of the fictive level of radiation were performed in Microsoft Excel, 2016.

## 3. Results

### 3.1. Baseline Characteristics

Our retrospective search yielded 153 patients who fulfilled the inclusion criteria, from which we randomly selected 40 patients. This sample size was chosen based on the assumption that it would capture the full spectrum of anatomical variation in this population, using a data saturation approach. This assumption was proven after the 20th patient, after which inclusion up to 40 patients confirmed this. The median age was 33 years (IQR 31–35) and the median BMI was 22.3 kg/m^2^ (IQR 21.0–23.6). Most patients were multiparous (52.5%). The majority of the patients had breast cup size C (25%) followed by cup size B (20%). The shortest distance from the ^125^I-seed to the fundal part of the uterus measured on PET-CT scans was 35.0 cm (mean: 39.6 cm, standard deviation (SD): 2.4 cm (95% CI: 34.9–44.3)).

In total, 19 patients were included in the outpatient clinic, of whom all gave consent to undergo the measurements. The average difference in distance between standing and sitting was 6.0 cm (95% CI 5.1–7.2).

### 3.2. Cumulative Fetal Dose

Table 1 and Table 2 show the cumulative dose absorbed by the fetus in the case of one implanted ^125^I-seed at consecutive gestational ages (GA), based on the ratio of one third sitting and two thirds standing or lying per day. The colors indicate exposure below 1 mSv (very light green), between 1 and 3 mSv (light green), between 3 and 5 mSv (darker green), and above 5 mSv (very dark green). The calculations were performed using the shortest measured initial distances between the ^125^I-seed and the fetus of 29.0 cm (35.0 cm minus 6.0 cm) (sitting) and 35.0 cm (standing/lying). The week (GA) of implantation of the ^125^I-seed during pregnancy is plotted against the week of surgery in GA. For example, when the ^125^I-seed is inserted at a GA of 20 weeks and the patient undergoes surgery at a GA of 24 weeks, the fetal cumulative dose of radiation is 0.2 mSv.

The main factor influencing the amount of radiation reaching the fetus is distance. Until a GA of 26 weeks, the exposure remains below 1 mSv because there is enough distance between the ^125^I-seed and the fetus. If implantation occurs at a GA of 32 weeks or beyond, the threshold of 1 mSv will be exceeded within one week because of the close proximity of the ^125^I-seed to the breast (Table 2). If surgery takes place in the same week as implantation from a GA of 32 weeks, the threshold will not be exceeded.

### 3.3. Translation to Clinical Setting

When primary surgery is indicated, the insertion of a ^125^I-seed for localization remains well below the threshold dose of tissue effects (100 mSv), which can already be considered safe. If surgery takes place up to a GA of 26 week, the exposure remains below 1 mSv, the worldwide accepted exposure for new born babies and members of the general public. Radiation exposure remains below 1 mSv if insertion and (surgical) removal take place within two weeks from a GA of 26 weeks, and within one week from a GA of 32 weeks (Table 2).

If neoadjuvant chemotherapy during the pregnancy is the preferred treatment option, the surgery will usually take place after the delivery. Chemotherapy can start, at the earliest, at 12 weeks of pregnancy, and usually takes at least 20 weeks. Since breast surgery is planned, on average, five to six weeks after chemotherapy, surgery in PrBC is performed after the delivery of the baby, with early induction if needed. To keep the dose to the fetus as low as possible in this described scenario, it is preferred that a non-radioactive marker is placed to mark the breast tumor before neoadjuvant chemotherapy, and the ^125^I-seed is placed postpartum.

If multiple ^125^I-seeds are needed for marking the breast tumor(s) or axillary lymph nodes, the cumulative dose (Table 1) can be multiplied by the number of seeds placed, as long the seeds have the same activity and are placed at the same time. The distance between the ^125^I-seed in the axilla and fetus is larger and usually has a lower intensity. For the three patients with the shortest distance (35.0 cm), the distance between the axilla and uterus was, at minimum, 45.0 cm, of which 5 cm went through lung tissue. The activity of an axillary ^125^I-seed was one MBq, equaling 0.0358 μGy m^2^/h per ^125^I-seed. This led to an additional contribution of 2.3% at maximum, meaning a dose of 0.5 mSv would become 0.512 mSv.

## 4. Discussion

This is, to our knowledge, the first study evaluating fetal exposure to a ^125^I-seed in the case of PrBC. The results of this study can function as a clinician’s guide for determining the preferred surgical localization technique and its timing without avoiding the use of a ^125^I-seed. It enables the clinician to reassure a pregnant patient with breast cancer, already subject to uncertainties and fear, that it is a safe method in her specific situation. For the clinician’s daily practice, it is also safer to follow the standard protocol without adaptations in the case of pregnancy. This helps minimize medical errors and enables the treating physician to offer the best medical care similar to that offered to non-pregnant breast cancer patients. Our study showed that ^125^I-seeds as a localization technique during BCS can be safely used during all stages of pregnancy. If surgery takes place before a GA of 26 weeks, the exposure always remains below 1 mSv.

The oncological outcome of PrBC has been previously studied and, in general, it is recommended to adhere to standard treatment protocols, whenever feasible [18]. One notable study by Amant et al. demonstrated comparable outcomes of disease-free survival and overall survival among women with breast cancer who received chemotherapy during pregnancy, as compared to non-pregnant young women [13].

The advantages of utilizing ^125^I-seeds as a localization technique for BCS in non-pregnant women have been widely acknowledged. Firstly, ^125^I-seeds have high accuracy and a low migration rate [24]. Secondly, the breast contour is known to be well preserved after wide local excision of the breast tumor, and we believe this also applies to PrBC. Additionally, re-excision rates are considerably lower when using ^125^I-seeds as a localization technique [28]. Moreover, when using the ^125^I-seeds for both marking as well as localization, any other additional techniques are abandoned [20]. Additionally, ^125^I-seeds are more cost-effective compared to wire-guided localization [41]. Given all the above-mentioned benefits, we recommend using ^125^I-seeds in PrBC whenever feasible. The addition of technetium-99m radioactivity in the sentinel node procedure on top of the ^125^I-seed is considered to be negligible and has been proven safe before [42].

The threshold of 1 mSv, the accepted dose for the public [32,35,37,43], is not reached in cases of implantation of a single ^125^I-seed during pregnancy up to a 26-week GA. After a 26-week GA, the fetal dose can be, at maximum 11.6 mSv, well below 100 mSv, above which fetal abnormalities can occur. If surgery takes place within two weeks of implantation from a GA of 26 weeks, and one week above 32 weeks, the dose remains below 1 mSv.

One should note that a very conservative safety threshold of 1 mSv was assumed in this study. Legally, there is no limit to the exposure of the mother and unborn child, as long as the medical procedure is justified [37]. Therefore, there is no binding reason to remain below this limit. However, the risks of not implanting a ^125^I-seed should be weighed against the additional risk of developing cancer.

For radiological exams such as CT-scans or PET/CT scans, doses to the fetus of 10 mSv are widely accepted. In this light, a higher dose to the fetus may be accepted, posing no practical limits to the use of one ^125^I-seed during any period in the pregnancy. According to the ICRP, exposure to the fetus should not exceed the limit of 100 mSv in order to prevent the risk of fetal abnormalities [34,44,45,46,47]. Furthermore, the lifetime risk of developing cancer increases by 0.01% per exposure to 1 mSv for a zero- to nine-year-old child. The maximum exposure of 11.6 mSv found in our study yields a theoretical increase in the risk of developing cancer of 0.12%. This is a small increase given that the lifetime risk of developing any type of cancer before the age of 75 is 35% for males and 33% for females [48,49]. Even with exceeding the threshold of 1 mSv, harmful effects on the fetus are not expected. Still, the general rule is to avoid exposing the public to a dose of more than 1 mSv.

The calculations performed in this study used assumptions that represent worst-case scenarios. First, the minimum amount of tissue between the fetus and the ^125^I-source was estimated to be 5 cm. This accounts for the (combined) thickness of the fetal environment, such as the myometrium, amniotic membranes and fluid, placenta, intra-abdominal organs, fatty tissue, and potentially breast tissue, that separates the fetus from the radioactive source. We believe that the minimum distance of 5 cm is a realistically safe (small) distance for regular pregnant women. In addition, the conversion factor of 0.35 mSv/mGy is rather high. Furthermore, we used the shortest distance between the ^125^I-seed and uterus found in the population of this study (35 cm) to cover a broad range of scenarios, but it is important to realize that anatomical variations in the population affect the radiation exposure of the fetus in utero. If, for example, a specific patient has, for some reason, a very short height or a very low BMI, the distance used could potentially be shorter, influencing the level of radiation reaching the fetus. Since we used rather extreme and safe distances, exposure will, in general, be lower. Van der Giessen estimated fetal dose from therapeutic irradiation of the breast as a function of GA [50]. For the measurements, distances from the fetus to the field center (breast) starting from 40 cm (8th week of pregnancy) and decreasing to 9 cm (36th week of pregnancy) were used. We showed comparable distances (mean: 40 cm), bearing in mind that we performed our calculations with the smallest distance found (35 cm).

In this study, physiological changes in the female body in reaction to pregnancy, such as the proliferation of mammary gland tissue, were not taken into account. This higher density resulted in even lower exposure to the fetus. In addition, the measurements were performed on anatomical landmarks and not on the actual growth of the uterus during a pregnancy monitored by imaging. The fact that the population consisted of non-pregnant women could be considered as a shortcoming of this study.

While previous studies have examined the psychological and emotional impacts of the diagnosis and treatment of cancer during pregnancy, stress specifically related to the use of ^125^I-seeds has not been mentioned [51]. After the implication of ^125^I-seeds in PrBC in clinical practice, this can be evaluated in time.

When implanting a ^125^I-seed in the case of PrBC, it is crucial to consider the potential overlap with the lactation period postpartum, especially towards the end of pregnancy [52,53]. During lactation, the threshold of 1 mSv could potentially be reached, as the neonate may be repeatedly and extensively exposed to radiation from the ^125^I-seeds through close proximity to the breast. Thus, future research should evaluate the levels of cumulative radiation reaching the fetus during lactation from both the ipsilateral and contralateral breast.

Based on the results of our study, the use of ^125^I-seeds as a localization technique for BCS during pregnancy is safe in most clinical scenarios. We recommend making an informed decision about the use of ^125^I-seeds during pregnancy mostly based on GA and the expected duration of implantation and time of surgery. On top of that, the physical aspects of the patient, such as height, BMI, and cup size, may play a part. Note that the amount of radiation due to the ^125^I-seeds may not be negligible and should therefore be added to the other possible sources of radiation during pregnancy.

## 5. Conclusions

In the case of PrBC, the use of ^125^I- seeds is safe and should no longer to be avoided. For primary surgery, ^125^I-seeds can be used safely in pregnancy since the radiation dosage remains well below the threshold of tissue effects of 100 mSv. Still, we propose keeping the dose to the fetus as low as possible, preferably below 1 mSv. Therefore, (surgical) removal must take place within two weeks from a GA of 26 weeks, and within one week from a GA of 32 weeks.

## Figures and Tables

**Table 1 cancers-15-03229-t001:** Fetal cumulative doses in consecutive gestational ages with one ^125^I-seed (0.358 uGy m^2^/h) in situ *.

Week of Surgery (GA)																							
**Week of implantation** ** ^125^ ** **I-seed (GA)**		**LMP**	**2**	**4**	**6**	**8**	**10**	**12**	**14**	**16**	**18**	**20**	**22**	**24**	**26**	**28**	**30**	**32**	**34**	**36**	**38**	**40**	**42**
	**LMP**	0.0	0.0	0.0	0.0	0.0	0.0	0.0	0.0	0.0	0.0	0.0	0.0	0.0	0.2	0.3	0.5	0.6	0.9	1.1	1.2	1.4	1.5
	**2**		0.0	0.0	0.0	0.0	0.0	0.0	0.0	0.0	0.0	0.0	0.0	0.1	0.2	0.4	0.6	0.7	1.0	1.2	1.4	1.6	1.7
	**4**			0.0	0.0	0.0	0.0	0.0	0.0	0.0	0.0	0.0	0.0	0.1	0.3	0.5	0.7	0.9	1.2	1.5	1.7	1.9	2.0
	**6**				0.0	0.0	0.0	0.0	0.0	0.0	0.0	0.0	0.0	0.1	0.3	0.5	0.8	1.0	1.4	1.7	2.0	2.2	2.4
	**8**					0.0	0.0	0.0	0.0	0.0	0.0	0.0	0.1	0.1	0.4	0.6	0.9	1.2	1.7	2.0	2.3	2.6	2.8
	**10**						0.0	0.0	0.0	0.0	0.0	0.0	0.1	0.1	0.5	0.7	1.1	1.4	1.9	2.4	2.8	3.1	3.3
	**12**							0.0	0.0	0.0	0.0	0.0	0.1	0.1	0.5	0.9	1.3	1.7	2.3	2.8	3.2	3.6	3.9
	**14**								0.0	0.0	0.0	0.0	0.1	0.1	0.6	1.0	1.5	2.0	2.7	3.3	3.8	4.3	4.6
	**16**									0.0	0.0	0.0	0.1	0.2	0.7	1.2	1.8	2.3	3.2	3.9	4.5	5.0	5.5
	**18**										0.0	0.0	0.1	0.2	0.8	1.4	2.1	2.7	3.7	4.6	5.3	5.9	6.4
	**20**											0.0	0.1	0.2	1.0	1.7	2.5	3.2	4.4	5.4	6.2	6.9	7.5
	**22**												0.0	0.1	1.0	1.8	2.8	3.6	5.0	6.2	7.2	8.0	8.7
	**24**													0.0	1.1	2.0	3.2	4.2	5.8	7.2	8.3	9.3	10.2
	**26**														0.0	1.1	2.5	3.6	5.5	7.1	8.5	9.7	10.7
	**28**															0.0	1.6	3.0	5.2	7.1	8.7	10.1	11.3
	**30**																0.0	1.6	4.2	6.5	8.4	10.0	11.4
	**32**																	0.0	3.1	5.8	8.0	9.9	11.6
	**34**																		0.0	3.1	5.8	8.0	9.9
	**36**																			0.0	3.1	5.8	8.0
	**38**																				0.0	3.1	5.8
	**40**																					0.0	3.1
	**42**																						0.0
**Cumulative Doses (mSv) Fetus**																							
**Legend:**				**<1.0 mSv**					**1.0–3.0 mSv**					**3.0–5.0 mSv**					**>5.0 mSv**				

* Calculations are performed based on the ratio of one third sitting and two third standing or lying per day. Abbreviations: GA, Gestational Age; ^125^I, Iodine-125; uGy, micro gray; m^2^/h, square meter per hour; LMP, Last Menstrual Period; mSv, millisievert.

**Table 2 cancers-15-03229-t002:** Fetal cumulative doses from a gestational age of 26 weeks with one ^125^I-seed (0.358 uGy m^2^/h) in situ *.

Week of Surgery (GA)																		
**Week of implantation** ** ^125^ ** **I-seed (GA)**		**26**	**27**	**28**	**29**	**30**	**31**	**32**	**33**	**34**	**35**	**36**	**37**	**38**	**39**	**40**	**41**	**42**
	**26**	0.0	0.6	1.1	1.8	2.5	3.1	3.6	4.6	5.5	6.4	7.1	7.9	8.5	9.1	9.7	10.2	10.7
	**27**		0.0	0.6	1.3	2.0	2.7	3.3	4.4	5.4	6.3	7.1	7.9	8.6	9.3	9.9	10.5	11.0
	**28**			0.0	0.8	1.6	2.3	3.0	4.1	5.2	6.2	7.1	8.0	8.7	9.5	10.1	10.7	11.3
	**29**				0.0	0.8	1.6	2.3	3.6	4.7	5.8	6.8	7.7	8.6	9.4	10.1	10.7	11.4
	**30**					0.0	0.8	1.6	3.0	4.2	5.4	6.5	7.5	8.4	9.3	10.0	10.8	11.4
	**31**						0.0	0.8	2.3	3.7	5.0	6.2	7.2	8.2	9.1	10.0	10.8	11.5
	**32**							0.0	1.6	3.1	4.5	5.8	6.9	8.0	9.0	9.9	10.8	11.6
	**33**								0.0	1.6	3.1	4.5	5.8	6.9	8.0	9.0	9.9	10.8
	**34**									0.0	1.6	3.1	4.5	5.8	6.9	8.0	9.0	9.9
	**35**										0.0	1.6	3.1	4.5	5.8	6.9	8.0	9.0
	**36**											0.0	1.6	3.1	4.5	5.8	6.9	8.0
	**37**												0.0	1.6	3.1	4.5	5.8	6.9
	**38**													0.0	1.6	3.1	4.5	5.8
	**39**														0.0	1.6	3.1	4.5
	**40**															0.0	1.6	3.1
	**41**																0.0	1.6
	**42**																	0.0
**Cumulative Doses (mSv) Fetus**																		
**Legend:**				**<1.0 mSv**				**1.0–3.0 mSv**				**3.0–5.0 mSv**				**>5.0 mSv**		

* Calculations are performed based on the ratio of one third sitting and two third standing or lying per day. Abbreviations: GA, Gestational Age; ^125^I, Iodine-125; uGy, micro gray; m^2^/h, square meter per hour; mSv, millisievert.

## Data Availability

The data presented in this study are available in this article.

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
