# Peer review of "The Safe Use of ^125^I-Seeds as a Localization Technique in Breast Cancer during Pregnancy"

_cancers, 2023, doi:10.3390/cancers15123229_

Round 1
Reviewer 1 Report
The article entitled “The Safe Use of 125I-seeds as Localization Technique in Breast Cancer During Pregnancy" clearly describes the objective of the study, which is to evaluate the safety of using 125I-seeds as a localization technique in patients with breast cancer during pregnancy. The developed model for calculating fetal exposure is mentioned and the factors considered in the calculation are explained. The presented results are concise and demonstrate that the fetal dose remains below levels considered safe. The conclusion effectively summarizes the findings of the study and emphasizes the safety of using 125I-seeds in PrBC. It would be helpful to mention the clinical implications or recommendations derived from the results.
According to the information provided by the authors, the study effectively addresses the safety of using 125I-seeds as a localization technique in breast cancer patients during pregnancy. However, to complement the study, the authors could consider investigating the following variables and provide a brief discussion:
In addition to evaluating the safety of the localization technique with 125I-seeds, it would be important to analyze long-term oncological outcomes, such as local recurrence rate, overall survival, and disease-free survival in patients treated with this technique. Furthermore, assessing long-term effects on the fetus, it would also be relevant to investigate long-term maternal outcomes, such as breast function, quality of life, and satisfaction with the treatment outcome. Additionally, considering the psychological and emotional impact of diagnosis and treatment, a psychological evaluation of patients undergoing breast-conserving surgery during pregnancy using different localization techniques would provide valuable insights. Moreover, conducting a detailed evaluation of the localization technique with 125I-seeds, including the accuracy of tumor localization, ease of use, and surgeon satisfaction with the procedure, would enhance the understanding of its effectiveness. Lastly, evaluating the long-term cosmetic results of breast-conserving surgery using the localization technique with 125I-seeds, considering the aesthetic appearance of the breast after the procedure and long-term patient satisfaction, would be beneficial.
Additionally, mentioning a cost-effectiveness analysis comparing the localization technique with 125I-seeds to alternative methods would provide valuable information on the economic implications of using this technique in the clinical setting.
Including a brief discussion of these variables in the discussion section would enrich the study by providing a more comprehensive understanding of the implications and outcomes associated with the use of 125I-seeds as a localization technique in breast cancer patients during pregnancy.
procedure and long-term patient satisfaction. By addressing these additional variables, the authors could obtain a more comprehensive and broader understanding of the results and effectiveness of the localization technique with 125I-seeds in breast cancer patients during pregnancy, which could provide valuable information for clinical decision-making and improve care for this population. The authors can enhance the scope and relevance of their study, providing a more comprehensive understanding of the implications and outcomes associated with the use of 125I-seeds as a localization technique in breast cancer patients during pregnancy and to mention a cost-effectiveness analysis comparing the localization technique with 125I-seeds to alternative methods would provide valuable information on the economic implications of using this technique in the clinical setting.
No comments
Author Response
Response to reviewer 1
Dear reviewer,
Thank you for reviewing our manuscript and providing valuable comments. Below, we have addressed each of your comments and made the necessary adjustments to our manuscript.
Point 1: The article entitled “The Safe Use of 125I-seeds as Localization Technique in Breast Cancer During Pregnancy" clearly describes the objective of the study, which is to evaluate the safety of using 125I-seeds as a localization technique in patients with breast cancer during pregnancy. The developed model for calculating fetal exposure is mentioned and the factors considered in the calculation are explained. The presented results are concise and demonstrate that the fetal dose remains below levels considered safe. The conclusion effectively summarizes the findings of the study and emphasizes the safety of using 125I-seeds in PrBC. It would be helpful to mention the clinical implications or recommendations derived from the results.
Response 1: We thank the reviewer for this comment. We add the clinical implications explicitly to the conclusion and changed the conclusion to:
“In case of PrBC, the use of 125I- seeds is safe and a combination no longer to be avoided. For primary surgery, 125I-seeds can be used safely in pregnancy since the radiation dosage remains well below the threshold of tissue effects of 100 mSv. Still, we propose to keep the dose to the fetus as low as possible, preferably below one mSv. Therefore, (surgical) removal must take place within two weeks from a GA of 26 weeks, and within one week from a GA of 32 weeks.”
According to the information provided by the authors, the study effectively addresses the safety of using 125I-seeds as a localization technique in breast cancer patients during pregnancy. However, to complement the study, the authors could consider investigating the following variables and provide a brief discussion:
Point 2: In addition to evaluating the safety of the localization technique with 125I-seeds, it would be important to analyze long-term oncological outcomes, such as local recurrence rate, overall survival, and disease-free survival in patients treated with this technique. Furthermore, assessing long-term effects on the fetus, it would also be relevant to investigate long-term maternal outcomes, such as breast function, quality of life, and satisfaction with the treatment outcome. Additionally, considering the psychological and emotional impact of diagnosis and treatment, a psychological evaluation of patients undergoing breast-conserving surgery during pregnancy using different localization techniques would provide valuable insights.
Response 2: The above mentioned points are indeed valid. These are partially related to the I-125 technique in general and not only in pregnancy. To study the long-term maternal outcomes using this technique and comparing it to another technique, many patients have to be included. This was not the focus of our study of I-125 in pregnancy. We therefore refer to a review/article specifically on this topic in non-pregnant women and state that based on these results, I-125 can be beneficial for pregnant women as wellhe oncologic outcomes of cancer in pregnancy have been described by members of our group before and a reference is added to the discussion. The psychological- and emotional impact of diagnosis and treatment of cancer in pregnancy has been studied before not specifically for stress related to the use of I-125. This is an excellent idea for a next study now we have showed that I-125 is safe to use in pregnancy. We added a sentence about this in the discussion.
“The oncological outcome of PrBC has been previously studied and in general it is recommended to adhere to standard treatment protocols, whenever feasible (18). One notable study by Amant et al., demonstrated comparable outcomes of disease-free survival and overall survival among women with breast cancer who received chemotherapy during pregnancy, as compared to non-pregnant young women (13). “
Regarding the psychological-and emotional impact of the diagnosis of cancer during pregnancy, we added this paragraph to the discussion:
‘While previous studies have examined the psychological- and emotional impact of diagnosis and treatment of cancer during pregnancy, stress-related to the use of I-125 seeds was not specifically addressed (51). Now we have showed that I-125 seeds can be safely used in PrBC, a study evaluating maternal stress related to I-125 seeds may be a next step. ‘
Point 3: Moreover, conducting a detailed evaluation of the localization technique with 125I-seeds, including the accuracy of tumor localization, ease of use, and surgeon satisfaction with the procedure, would enhance the understanding of its effectiveness. Lastly, evaluating the long-term cosmetic results of breast-conserving surgery using the localization technique with 125I-seeds, considering the aesthetic appearance of the breast after the procedure and long-term patient satisfaction, would be beneficial.
Response 3: Regarding the oncological safety, Taylor et al., found that re-excision was considerably lower when using an I-125 seed as localization technique compared to the wire-guided technique, with a re-excision rate of 13.9% versus 18.9% respectively (p-value = 0.019). We added this study to the introduction:
“The use of Iodine-125 (125I) seeds as localization technique has grown in popularity in the past decade due to its high accuracy and low migration rate and has rapidly replaced the former golden-standard, the wire-guided localization technique (19, 22-27). Moreover, re-excision rates are considerably lower when using 125I-seeds as localization technique.”
And we added this paragraph to the discussion:
“The advantages of utilizing I-125 seeds as localization technique for BCS in non-pregnant women have been widely acknowledged. Firstly, I-125 seeds have a high accuracy and low migration rate (24). Secondly, the breast contour is known to be well preserved after wide local excision of the breast tumor and we believe this also applies for PrBC. Additionally, re-excision rates are considerably lower when using 125I-seeds as localization technique (28). The simplicity of surgical scheduling is another advantage, as I-125 seed localization eliminated the requirement for additional radiological localization just before surgery (20). Moreover, I-125 seeds are more cost-effective compared to wire guided localization (41). Given these benefits of using I-125 seeds and the knowledge gained from this study that it can be used safely during pregnancy, we recommend the utilization of I-125 seeds as localization technique in PrBC. “

Reviewer 2 Report
I think the paper is interesting and well written and can be accepted in the present form.
I would suggest to describe in more detail the rationale of using radioactive localisation techniques in breast cancer surgery.
I also think that citing the paper by Gentilini et al. (Safety of sentinel node biopsy in pregnant patients with breast cancer PMID: 15319240) may be of interest since they also describe a model to evaluate foetal exposure to raditation during a procedure involving the use of radioactive material.
Author Response
Response to reviewer 2
Dear reviewer,
Thank you for reviewing our manuscript and providing valuable comments. Below, we have addressed each of your comments and made the necessary adjustments to our manuscript
Point 1: I think the paper is interesting and well written and can be accepted in the present form.
I would suggest to describe in more detail the rationale of using radioactive localisation techniques in breast cancer surgery.
Response 1: We thank the reviewer for this valuable comment.
We added the following to the introduction:
“The use of Iodine-125 (125I) seeds as localization technique has grown in popularity in the past decade due to its high accuracy and low migration rate and has rapidly replaced the former golden-standard, the wire-guided localization technique (19, 22-27). Moreover, re-excision rates are considerably lower when using 125I-seeds as localization technique.”
And we added this paragraph to the discussion:
“The advantages of utilizing I-125 seeds as localization technique for BCS in non-pregnant women have been widely acknowledged. Firstly, I-125 seeds have a high accuracy and low migration rate (24). Secondly, the breast contour is known to be well preserved after wide local excision of the breast tumor and we believe this also applies for PrBC. Additionally, re-excision rates are considerably lower when using 125I-seeds as localization technique (28). The simplicity of surgical scheduling is another advantage, as I-125 seed localization eliminated the requirement for additional radiological localization just before surgery (20). Moreover, I-125 seeds are more cost-effective compared to wire guided localization (41). Given these benefits of using I-125 seeds and the knowledge gained from this study that it can be used safely during pregnancy, we recommend the utilization of I-125 seeds as localization technique in PrBC. “
Point 2: I also think that citing the paper by Gentilini et al. (Safety of sentinel node biopsy in pregnant patients with breast cancer PMID: 15319240) may be of interest since they also describe a model to evaluate foetal exposure to raditation during a procedure involving the use of radioactive material.
Response 2: We thank the reviewer for this valuable addition. We added the following to the discussion:
“The addition of the 99m-Technetium radioactivity for the sentinel node procedure on top of the I-125 seed is considered to be negligible and has been proven safe before (42). “